# Immune Pressure on Polymorphous Influenza B Populations Results in Diverse Hemagglutinin Escape Mutants and Lineage Switching

**DOI:** 10.3390/vaccines8010125

**Published:** 2020-03-11

**Authors:** Ewan P. Plant, Hasmik Manukyan, Jose L. Sanchez, Majid Laassri, Zhiping Ye

**Affiliations:** 1Division of Viral Products, Center for Biologics Evaluation and Research, US Food and Drug Administration, Silver Spring, MD 20993, USA; Hasmik.Manukyan@fda.hhs.gov (H.M.); Majid.Laassri@fda.hhs.gov (M.L.); zhiping.ye@fda.hhs.gov (Z.Y.); 2Armed Forces Health Surveillance Branch, Public Health Division, Assistant Director for Combat Support (AD-CS), Defense Health Agency, Silver Spring, MD 20904, USA; jose.l.sanchez76.civ@mail.mil

**Keywords:** influenza B, hemagglutinin, lineage, quasispecies, next generation sequencing

## Abstract

Mutations arise in the genomes of progeny viruses during infection. Mutations that occur in epitopes targeted by host antibodies allow the progeny virus to escape the host adaptive, B-cell mediated antibody immune response. Major epitopes have been identified in influenza B virus (IBV) hemagglutinin (HA) protein. However, IBV strains maintain a seasonal presence in the human population and changes in IBV genomes in response to immune pressure are not well characterized. There are two lineages of IBV that have circulated in the human population since the 1980s, B-Victoria and B-Yamagata. It is hypothesized that early exposure to one influenza subtype leads to immunodominance. Subsequent seasonal vaccination or exposure to new subtypes may modify subsequent immune responses, which, in turn, results in selection of escape mutations in the viral genome. Here we show that while some mutations do occur in known epitopes suggesting antibody escape, many mutations occur in other parts of the HA protein. Analysis of mutations outside of the known epitopes revealed that these mutations occurred at the same amino acid position in viruses from each of the two IBV lineages. Interestingly, where the amino acid sequence differed between viruses from each lineage, reciprocal amino acid changes were observed. That is, the virus from the Yamagata lineage become more like the Victoria lineage virus and vice versa. Our results suggest that some IBV HA sequences are constrained to specific amino acid codons when viruses are cultured in the presence of antibodies. Some changes to the known antigenic regions may also be restricted in a lineage-dependent manner. Questions remain regarding the mechanisms underlying these results. The presence of amino acid residues that are constrained within the HA may provide a new target for universal vaccines for IBV.

## 1. Introduction

Influenza B virus (IBV) belongs to the Orthomyxoviridae virus family and causes significant morbidity and mortality each year [1,2]. Humans are the primary host of this segmented, negative-strand RNA virus. The major surface protein, hemagglutinin (HA), is encoded by the fourth of eight segments. HA is involved in receptor binding and membrane fusion and is one of the major antigenic proteins targeted by the host immune system [3].

IBVs evolved into two lineages that diverged in the 1970s [4]. The lineages are named after the B/Victoria/2/1987 and B/Yamagata/16/1988 strains [2]. Before 1985, the precursor to the Yamagata lineage circulated [5]. The Victoria lineage circulated globally in the late 1980s and then the Yamagata lineage dominated in the 1990s [2,4].

Like other Orthomyxoviridae, IBVs encode an RNA polymerase that is used for replication. Errors during replication of influenza viruses result in quasispecies [6]. Not all the viruses are viable and genetic bottlenecks occur when the viable viruses infect new hosts [7]. The virus keeps evolving and antigenically distinct viruses emerge each year. It has been shown for an A(H3N2) influenza A virus (IAV) that only one mutation was required for the virus to escape the host adaptive, B-cell mediated, antibody immune responses [8]. Thus, updates to the seasonal vaccine formulation are frequently required in order to encourage the production of antibodies specific to the circulating viral strains.

The goal of many universal influenza vaccines is to generate antibodies that are protective against multiple strains of influenza. Much of the immune response is directed toward less well conserved immunodominant epitopes, and one of the major challenges in developing universal vaccines is directing the immune response toward the more conserved sites [9]. However, much of this knowledge is based on IAV studies and the responses to IAV and IBV strains differ [10]. Infection, or vaccination with live attenuated influenza viruses, may play a role in the development of cross-reactive antibodies. Broadly neutralizing antibodies are selected at viral replication sites [11] and animal models have shown that sequential infection with IAVs can generate antibodies that are protective against other IAV strains [12]. There is some evidence for cross-reactive responses toward IBVs in animals [13]; however, it is not known how much of this effect is due to B-cell mediated adaptive versus T-cell mediated innate immunity.

There is evidence that broadly protective antibodies toward IBVs are produced in humans. First, IBVs predominantly affect children and adolescents 18 years of age or younger [14,15]. This most likely indicates that some form of long-lasting protection is acquired after individuals are infected before adulthood. Second, human monoclonal antibodies that protect mice against lethal challenge from IBV from both lineages have been described [16,17]. Furthermore, stimulation of peripheral blood mononuclear cells from healthy donors with IBV from either lineage results in IBV-specific CD8+ T lymphocytes that cross react with viruses from both lineages including drift variants [18]. Finally, antibodies that target noncanonical IBV epitopes increase with age [3].

Cross-reactive antibodies can be detected after either infection or vaccination. Primary human IBV infection has been shown to induce cross-lineage hemagglutinin stalk-specific antibodies that mediate antibody-dependent cell-mediated cytotoxicity [19]. Broadly protective antibodies generated by vaccination have been shown to be relatively abundant in sera [20]. Antibodies arising from a B/Yamagata/16/1988 vaccination in adults were shown to recognize B/Victoria/2/1987 in the early 1990s [21]. Broadly reactive antibodies that act both within and between lineages can also be elicited after vaccination in an older population [22]. However, it was demonstrated that children that had not been exposed to the B/Victoria/2/1987 strain did not produce cross reactive antibodies after vaccination with the B/Yamagata/16/1988 strain [23], suggesting that exposure to strains from both lineages may be necessary for the production of broadly protective antibodies. More recently it has been shown that exposure to Yamagata strains seems to result in an enhanced cross-reactive antibody response when compared to exposure to Victoria strains [17,24].

Here we investigate the changes to the virus HA coding sequence after growth in the presence of human sera from subjects vaccinated in two sequential seasons with Victoria lineage antigens. Sera from historical samples was used with contemporary viruses. We hypothesize that sera from vaccinated adults will contain cross-reactive antibodies that will drive escape mutations in viruses from both IBV lineages.

## 2. Materials and Methods

### 2.1. Sera Selection

Influenza B escape viruses were generated by propagating two viruses (one each from the Victoria and Yamagata lineages) in the presence of sera collected 2–6 weeks post vaccination. The sera were from military personnel that had received at least two vaccinations. Three sera samples were collected in 2003 from recipients of trivalent inactivated influenza vaccines (IVV) containing the B/Hong Kong/330/01 antigen, and three sera samples were collected in 2011 from recipients of a live attenuated influenza vaccine in 2010 and an inactivated influenza vaccine in 2011, both containing the B/Brisbane/60/2008 antigen [25]. Sera from individuals with different antibody titers (10, 40, and 320) to the respective vaccine antigens were selected to test whether the titers were important for antibody escape. The sera were treated with receptor destroying enzyme as described below and subjected to a heat inactivation step that inactivates both the enzyme and complement.

### 2.2. Preparation of Escape Mutants

An equal volume (50 µL) of virus and serum were incubated together at room temperature for 30 min. The initial virus HA titers used were 4 for the Yamagata lineage B/Phuket/3073/2013 (B/PK) strain and 32 for the Victoria lineage B/Colorado/06/2017 (B/CO) strain. Both viruses are cell culture propagated isolates obtained from the CDC. We used more B/CO because we anticipated that subjects vaccinated two years in a row with a Victoria lineage antigen would have more antibodies toward that lineage. Three different dilutions of receptor-destroying enzyme treated serum were used, 1:10, 1:40, and 1:160. The mixture was used to inoculate MDCK cells (from Dr. Hang Xie, [26]) in a 12-well plate. After one hour the inoculum was removed and replaced with culture media. Hemagglutination assays were performed 24–48 h post infection. Media from the well with the most concentrated sera with detectable hemagglutination titers was diluted to an HA titer of 4 and the process repeated 1–4 times using less dilute sera (up to 1:10).

### 2.3. RNA Library Preparation, Illumina Sequencing and Data Analysis

RNA was extracted from the media using the QIAamp Viral RNA Mini Kit (Qiagen, Germantown, MD, USA) following the manufacturer’s instructions. The RNA library was prepared using the NEBNext Ultra RNA Library Prep Kit for Illumina (New England BioLabs, Ipswich, MA, USA). Fragmentation and priming were performed in one reaction using the fragmentation buffer provided in the kit, and the fragmentation conditions were optimized to result in about 300 nt RNA fragments. The first (cDNA) and second strands of DNA were synthesized according to the manufacturer’s protocol. The resulting DNA fragments were ligated to Illumina paired end (PE) adaptors, then amplified using 12 cycles of PCR with multiplex indexed primers and purified by magnetic beads (Agencourt AMPure PCR purification system, Beckman Coulter, Brea, CA, USA). After analyzing the libraries for the size and quality with BioAnalyzer (Agilent Technologies, Santa Clara, CA, USA), deep sequencing was performed using MiSeq (Illumina, San Diego, CA, USA) producing 250 nt paired end reads. The raw sequencing reads were analyzed by an inhouse developed specialized platform High-performance Integrated Virtual Environment (HIVE) [27,28]. Raw sequence data is available at https://hive.biochemistry.gwu.edu/review/Immune%20pressure%20on%20polymorphous%20influenza%20B%20populations%20results%20in%20diverse%20escape%20mutants%20and%20lineage%20switching (additional descriptive information is provided in the Appendix A).

The RNA sequences of B/Utah/09/2014 and B/Colorado/06/2017 viruses were downloaded from NCBI GenBank and used as genome references for HIVE alignment and mutations profiling of sequencing reads generated from B/Phuket/3073/2014 (B/PK) and B/Colorado/06/2017 (B/CO) viruses respectively.

### 2.4. Hemagglutination Inhibition Assays

Hemagglutination Inhibition (HAI) titers were determined as previously described [25]. Briefly, each volume of sera was treated with 3 volumes of receptor destroying enzyme (Accurate Chemical, Westbury, NY, USA) at 37 °C overnight, which was then inactivated by heating in a 56 °C water bath for 30 min. Six volumes of phosphate buffered saline were added to make the initial sera concentration of 1:10. This was used for both preparation of escape viruses and HAI assays. Serial dilutions of RDE treated serum were prepared and incubated with an equal volume containing 4 HA units of virus per well. No virus wells were included for each sera and the plates incubated at room temperature for 30 min. Freshly diluted turkey red blood cells were added to all wells and the plates incubated for an additional hour. After incubation the reciprocal of the last dilution that completely inhibited agglutination was recorded. Four technical replicates were performed. Results with no inhibition were recorded with a value of 5 and geometric mean titers were calculated from the log values.

## 3. Results

Two viruses, B/Phuket/3073/2013 (B/PK) and B/Colorado/06/2017 (B/CO), were independently passaged in MDCK cells in the presence of post-vaccination sera from six individuals who received seasonal influenza vaccine. Three of the sera samples were from service members who received IIV vaccines containing the B/Hong Kong/330/01 antigen (HK330) in the 2002/03 and 2003/04 seasons. The other three samples were from service members who received a LAIV in the 2010/11 season and an IVV in the 2011/12 season, each vaccine containing the B/Brisbane/60/2008 antigen (BR60) (Table 1 and Table 2). The HAI titers to the Victoria lineage vaccine antigens (taken from [25]) are shown in Table 1 along with titers toward the contemporary B/CO virus. The titers toward B/CO were lower but there was a correlation with the vaccine antigen titers. Similarly, titers toward the Yamagata lineage B/PK virus were higher with sera containing higher titers toward the vaccine antigens (Table 2).

Viral RNA was extracted from culture supernatant and subjected to Illumina sequencing (Next Generation Sequencing; NGS). HA consensus sequences were determined for each virus, translated into protein sequences, and compared (Figure 1). The sera used in this study were collected in 2003 and 2011, before the circulation of the viruses used to generate escape mutants. The Victoria lineage B/CO virus was recommended for inclusion in the 2018–2019 and 2019–2020 northern hemisphere vaccines and is antigenically distinct from the vaccine antigens. The Yamagata lineage B/PK strain was recommended for the 2015–2016 northern hemisphere vaccine and the vaccine recipients were not vaccinated with a Yamagata lineage antigen in the seasons referenced. Thus, the antibodies present in the sera are not expected to be antigenically well matched to the viruses used to generate the escape mutants. When the B/CO virus was cultured in the presence of the sera only two of the escape viruses had changes in the HA amino acid sequence (CO/M74 and CO/F16; Figure 1). The CO/M74 virus had changes in the antigenic 120-, 150-, and 190-loops. The CO/F16 virus had a single change (G141R) in the antigenic 150-loop. Additionally, there were many changes throughout the HA sequence at positions where the CO/WT and PK/WT strains are not homologous. Notably, the sera from individuals with high (320) HAI titers toward the IBV antigen included in the vaccines they received (M74, B/Hong Kong/330/01 and F16, B/Brisbane/60/2008 respectively; Table 1) resulted in the escape viruses with the most changes to the HA gene.

When the Yamagata lineage B/PK virus was cultured in the presence of sera, the HA consensus sequences of all six escape viruses differed from the HA amino acid consensus sequence for the parental virus. Culture in the presence of sera with medium or high HAI titers (40 and 320) toward the B/Hong Kong/330/01 antigen, or sera with a low titer (10) toward the B/Brisbane/60/2008 antigen yielded viruses with changes in the antigenic 120- and 150-loops (escape viruses PK/M50, PK/M74 and PK/M09 respectively; Figure 1). Culture of the B/PK virus in the presence of sera with a low HAI titer (10) toward the B/Hong Kong/330/01 antigen, or sera with medium and high titers (40 and 320) toward the B/Brisbane/60/2008 antigen yielded viruses with changes in positions where the HA sequences of the B/PK and B/CO viruses differ (escape viruses PK/F09, PK/M56, and PK/F16 respectively; Figure 1). The PK/F16 virus also had a change in the 150-loop (G141R) and a deletion in the 160-loop.

The parental viruses, B/PK and B/CO, differ at 37/575 (584) amino acid positions in the HA protein and B/PK has one more residue in the 160-loop (Figure 1). These positions are solvent accessible [29] and, interestingly, many of the changes observed made the B/PK escape viruses more like B/CO and vice versa. Thirty-four of the forty-two changes observed in the CO/M74 virus were at amino acid positions where B/CO and B/PK differ. Similarly, many of the changes (33/35, 3/3, and 38/40 for PK/F09, PK/M56, and PK/F16 respectively) made the Yamagata lineage escape viruses more like B/CO.

The changes in the antigenic loops followed different patterns. Some changes were common in the escape viruses even though the viruses were from different lineages and derived using different sera. For example, identical changes to the 120- and 150-loops were observed for the CO/M74, PK/M74, PK/M50, and PK/M09 viruses (Figure 1). Some changes were specific to a particular serum. At position 129 (HA1 numbering) there was a glycine to lysine change in CO/M74. Similarly, the PK/M74 escape virus also changed to a lysine at this position, albeit from an aspartic acid. Substitutions at this position have been reported for both IBV lineages indicating that it is a variable position [30]. At position 141 (HA1 numbering) there was a glycine to arginine change in both the CO/F16 and PK/F16 viruses. Finally, some changes in the antigenic loops made the escape virus from one lineage more like the other lineage (changes to the 150-, 160- and 190-loops in CO/M74, PK/F09 and PK/F16, and changes to the 120-loop in PK/M56 and PK/F16 for example).

The differences in the 160 loop help define the two IBV lineages [5]. Changes to the amino acids between 163 and 167 (HA1 numbering) in Yamagata and Victoria lineage viruses have been reported to contribute to antigenic change [31]. Glycosylation changes the structure of a protein and has been shown to affect growth or antigenicity of influenza viruses. In the 160-loop there are potential glycosylation sites in both the B/PK and B/CO viruses. In B/CO there is a 163NKT sequence and in B/PK there is a 165NAT sequence. The CO/M74 escape virus had the changes N163Y and T165N, which remove the potential glycosylation site at position 163–165 and create a new one at position 165–167. The PK/F09 and PK/F16 escape viruses had changes Y163N and N165T removing the potential glycosylation site at position 165–167 and creating one a position 163–165. Govorkova et al. reported the loss of a glycosylation site at position 194 after virus was passaged in baby hamster kidney cells [32]. Potential glycosylation sites emerged in the escape viruses CO/M74 and PK/F09 at position 194 after culture in the presence of sera. These results provide further support that the 160-loop helps define the lineages, but it remains unclear what role the potential glycosylation sites have.

The changes observed in the HA coding regions of the escape viruses are expected to result in viruses that bind fewer antibodies from the sera used to generate them. We measured HAI titers of the sera toward the wild-type B/Colorado/06/2017 virus and two of the escape viruses, CO/M74 and CO/F16. Due to limited sera samples we restricted the experiments to the viruses with the most divergent sequences. The titers to the CO/WT virus were low overall but a trend for higher titers for the sera from the subjects that had the highest titers to the IBV vaccine antigen (Figure 2A). We measured HAI titers of the sera toward the wild-type B/Phuket/3073/2013 virus and two of the escape viruses, PK/F09 and PK/M09. The titers to the PK/WT virus were higher for the sera from the subjects that had the highest titers to the IBV vaccine antigen (Figure 2B). The titers toward the PK/F09 and PK/M09 viruses were lower than those for PK/WT in all instances except for the F16 sera toward the PK/M09 virus. The PK/F09 virus had changes that made the backbone more like the Victoria lineage viruses and sera titers were lower for this virus than the PK/M09 virus which had changes to the antigenic loops. Our results show that the HAI titers are generally lower with the escape mutants indicating that fewer antibodies in the sera are able to bind. The notable exceptions are the PK/M09 and PK/M56 viruses in the F16 sera indicating that the F16 sera (which had high titers toward the vaccine antigen) has an antibody repertoire that differs from the M09 and M56 sera.

Viruses exist as a population and NGS allows us to investigate the sequences of these populations in greater depth. Specifically, we looked for the presence of variants that had emerged in the escape viruses in the parental strains. Sequence variants present at levels greater than 5% of the total number of reads were identified at each position in the HA gene (Table 3). Of the 42 mutations identified at levels greater than 50% in the Victoria-lineage escape viruses, only two of those variants were detected at levels above 5% of the sequence reads in the parental CO/WT population. In contrast, 42 of the 45 variants in the Yamagata-lineage escape viruses were present at levels greater than 5% in the PK/WT population (Table 3). This indicates that the PK/WT virus existed as a more diverse population.

We compared the growth of the escape viruses. MDCK cells were inoculated with a low multiplicity of infection (MOI = 0.001). The inoculum was removed after one hour and replaced with media. Aliquots were taken at different timepoint and titers determined using the plaque assay on MDCK cells. Titers were detected at the 0-h timepoint indicating that some inoculum remained that included unbound virus. The initial growth rates for the escape viruses paralleled those of the wild-type viruses in most instances regardless of the original titer. The CO/F16 and PK/F16 viruses had slightly lower titers after 24 h but all viruses reached titers comparable to the parental strains after 72-h (Figure 3). This indicates that the changes to the viral population did not affect growth.

## 4. Discussion

It is expected that antibodies directed towards a specific epitope would govern the evolution of escape mutants. However, it is not clear what effect polyclonal and cross-reactive antibodies generated after multiple vaccinations or exposures have on IBV evolution. There is evidence to suggest that receiving vaccines with the same HA stalk region and different head regions may increase the antibodies that target the more conserved regions of the HA protein [33]. Other work indicates that antibodies generated towards viruses encountered early in life may be expanded and comprise a large portion of the influenza-specific antibodies [34]. These antibodies are focused on specific antigenic sites [35] and escape from such narrowly focused human serum antibodies has been demonstrated in ovo [36]. In this work, we observed the emergence of several Yamagata lineage B/PK escape viruses with similar mutations after amplification in the presence of antibodies from different people who received vaccines containing antiquated Victoria lineage antigens. Although the vaccines were not antigenically matched to the viruses used in this work, some of the escape mutations described here are in previously described antigenic loops, supporting the notion that these loops are targeted by the immune system. However, many changes in the escape viruses were in other solvent accessible areas of the HA, suggesting a greater breadth of antibody susceptible areas on IBV HA. Almost all the mutations observed in the HA1 region correspond to the positions of high entropy [37].

We were able to assess the diversity of the virus populations from both the parental strains and the escape viruses using deep sequencing. Next Generation Sequencing (NGS) revealed that the B/Phuket/3073/2013 virus population used in this work was more polymorphous than the B/Colorado/06/2017 virus population. Our experimental design likely resulted in a large portion of the virus population being transferred in each round of passage. This would be expected to result in a more diverse quasispecies population compared to passage using a low multiplicity of infection [38]. In line with this, our analysis revealed more codon changes (at a frequency >50%) in the B/PK Yamagata lineage escape viruses than in the B/CO Victoria lineage escape viruses. These in vitro results are massively more pronounced than reports based on analyses of native virus samples. An analysis of IAVs found that bottlenecks restricted the amount of diversity that was shared after a virus was transmitted from one individual to another [7]. Even less diversity was found in IBV transmitted from one individual to another, although the samples with the highest number of intrahost single nucleotide variants were excluded from the analysis because it was postulated that they could be due to a coinfection [39]. Sampling from the human population indicates that the Victoria lineage IBVs have frequent bottlenecks due to serial replacement of circulating strains [15]. In contrast to the serial replacement of Victoria strains, multiple Yamagata clades co-circulated for extended periods of time from 2002 to 2012 and had greater genetic diversity [15]. This correlates with the in vitro experiments in this work which revealed that a greater number of mutants emerged from the more polymorphous Yamagata strain than was observed with the less polymorphous Victoria strain. Vijaykrishna et al. also reported that 15% of the Victoria lineage IBVs acquired all the internal gene segments from the Yamagata lineage [15]. Our work raises the possibility that these could have been Yamagata strains with HA genes mutated such that they fell into the Victoria lineage.

It is apparent from our in vitro analysis that the availability of genetic variants has a role in the emergence of escape mutants. It has been shown that viruses that are less fit can become overrepresented in the population through coinfection [40]. Such viruses have been observed in influenza studies and it has been shown that several different types of influenza viruses with incomplete genomes can be propagated with complementary viruses [41,42,43]. It has also been shown that virus yield is affected by the diversity produced after bottleneck events [44]. These works support our assertion that the existing diversity in the virus population used in this study likely played a role in expanding the diversity of the population of escape viruses.

The emergence of identical mutations across different experiments requires further investigation. The rise of identical polymorphisms after selection with different sera suggests that the different sera applied similar selective pressure. That the changes arose in the same positions in the presence of sera from several different individuals suggests that either similar antibodies were present in the various sera or that the changes to the protein are constrained to certain areas. The mutations were not equally distributed across the HA gene and that they arose en masse was somewhat unexpected. This indicates that the viruses emerged as much as a result of selection from the existing quasispecies pool as through mutation during replication. Recent analysis of negative-strand RNA virus genomes has found that their codon usage bias does not correlate well with the host codon bias because of constraints in viral RNA transcription [45]. It is possible that codon mutations, while limited by transcription and translation requirements, are also constrained by other features such as genomic RNA structures [46]. Unfortunately, the sequencing employed for this work does not allow for detailed analysis of the linkage of the various mutations across the whole genomic segment. However, we observed that while the frequency of each mutation differed among the viruses, pairs of mutations in the same codon were often observed at the same frequency even when only one mutation was required to encode a different amino acid (Table 3).

We looked to see if there was an association between where the number of changes that occurred in the escape viruses’ HA sequence and the antibody titers in the sera towards the vaccine antigen. There were specific types of changes that occurred with certain sera. We observed that the only two B/CO escape viruses with changes in the HA amino acid sequences were from sera with the highest titer to the IBV antigen included in the vaccines (M74 and F16). In addition, both escape viruses with the G141R mutation were generated with the sera with the highest titer toward the B/Brisbane/60/2008 (F16). Antibody levels peak two weeks after vaccination, and because the sera used in this study were collected 2–6 weeks after vaccination, there could possibly be a loss in the quantity and quality of the influenza antibodies in sera drawn in the latter portion of the window. For each round of viral amplification, the least diluted sera sample we could obtain virus from was used, which differed between sera samples. However, there was no relationship between the actual number of changes to the HA coding sequence and the titers toward the vaccine antigens. This may be due to differences in the ratios of antibody to virus we used.

We hypothesized that viruses would require multiple mutations to escape the polyclonal antibodies present in human sera. However, we did not expect to see identical changes in viruses from different lineages or identical changes with sera from recipients of different vaccines. There are explanations for these observations. For example, antibodies may target epitopes that are conserved between the two lineages and repeated vaccination (or exposure) may train the immune response so that it contains targets more cross-lineage epitopes. We observed identical changes in both the Victoria and Yamagata lineage viruses when grown in the same sera. The two escape viruses recovered after growth in the presence of the F16 sera both had the same G141R change (HA1 numbering) in the 150 loop. This change has also been reported in other escape mutants [17]. Virus propagation in the M74 sera resulted in identical changes to the 120 and 150 antigenic loops for the Yamagata (PK/M74) and Victoria (CO/M74) lineage viruses. However, the CO/M74 virus also had changes throughout the HA that made it more a Yamagata lineage virus. This suggests that the antibodies in the M74 sera, in addition to selection against the epitopes in the 120 and 150 antigenic loops, strongly inhibited the growth of the virus with a Victoria-like sequence. Supporting this idea, the M74 sera was from a person with a high HAI titer (320) toward the older B/Hong Kong/330/2001 antigen and a slightly higher titer toward B/CO (22; Table 1) compared to the other escape viruses. Changes to the solvent accessible regions of HA were not observed in the other B/CO escape viruses suggesting the M74 serum differed from the other sera in this regard.

There are certain motifs that are more frequently associated with each lineage. The amino acids in the 120-loop are proximal to some solvent accessible residues [47]. We observed that the N126D, G129K and I137L changes occurred in the context of D56, V73 and T75 Yamagata-like residues but not K56, T73, and K75 Victoria-like residues. The ^148^NGN^150^ motif is more commonly found in Victoria lineage strains [47]. The motif was also present in both the parental viruses used in this study. The ^148^NGN^150^ motif changed to ^148^SKI^150^, which is more commonly found in Yamagata lineage strains, in four of the escape viruses (PK/M09, PK/M50, PK/M74 and CO/M74) suggesting a strong impact from antibodies targeting the ^148^NGN^150^ motif.

Binding the sialic acid moiety facilitates virus attachment and structural analyses have identified residues that form hydrogen bonds with the sialic acid [47]. Two of the escape viruses (PK/F16 and CO/F16) have a mutation at one of these residues (G141R, HA1 numbering) and four escape viruses (PK/M09, PK/M50, PK/M74 and CO/M74) have a mutation (K136R) at another. The G141R change occurred in the context of Victoria-like sequences and the K136R change occurred in the context of Yamagata-like lineages. Five of these viruses had lower titers at the 24-h time point in the growth curves suggesting some loss in fitness (Figure 3), but all viruses reached similar titers by 72 h.

If the immune pressure placed on the input virus strains were due to antibodies specific to one lineage, we would expect changes in only the virus from that lineage, yet we observed changes at the same positions in both virus lineages when either M74 or F16 sera was present. This suggests the presence of antibodies that target both lineages. However, in addition to the changes common to both viruses within the antigenic 120- and 150-loop, there were additional changes to the solvent-exposed residues such that each escape virus grouped in the same lineage (Victoria lineage for the PK/F16 and CO/F16 escape viruses and Yamagata lineage for the PK/M74 and CO/M74 escape viruses). This may indicate that changes to known antigenic sites are more stable in the context of a particular lineage.

We asked if changes that made one virus appear to belong to another lineage were due to the sera or were related to other experimental conditions such as culture growth conditions or host cell type. Identical changes to the HA sequences were observed for some of the Yamagata escape mutants recovered after growth in different sera and opposing changes were observed with different serum for one of the Victoria escape mutants. This effect was not observed for all the different sera used, nor was it common to sera only from 2003 or 2011. As the same cells, media, and growth conditions were used we speculate that it is more likely that the changes are due to immune pressure resulting from growth in the presence of specific sera rather than growth conditions or host cell factors. There is evidence that IBV predominantly affects children 18 years or younger [14,15]. This suggests that as people age, they acquire antibodies to prevent infection by IBV from either lineage. It has also been shown that influenza vaccination in the older population is able to elicit broadly reactive antibodies both within and between lineages [22]. The sera used in this study were from young adults (ages 19–21) indicating that antibodies able to suppress IBVs from the same, or even different, lineages are present in young vaccinated adults.

We considered whether cross-reactive antibodies required a particular exposure history prior to vaccination. After the IBVs split into two distinct lineages there were some studies that investigated cross-reactive IBV antibodies. Levandowski et al. demonstrated that vaccination with the B/Yamagata/16/1988 strain generated antibodies that recognized B/Victoria/2/1987 [21]. Subsequently, it was shown that exposure to Yamagata lineage strains more often resulted in cross-reactive antibodies than exposure to Victoria lineage strains [17,24]. Analysis of antibodies from an individual vaccinated twice with a seasonal trivalent vaccine containing the Yamagata lineage B/Massachusetts/2/2012 antigen revealed antibodies that could bind to both lineages or Yamagata lineage viruses alone, but no antibodies that bound only Victoria lineage viruses [48]. The sera used in this study were from young adults that were children in the 1980s and 1990s and all were vaccinated two years in a row with Victoria lineage antigens. Individuals born in the early 1980s were likely exposed to both Victoria and Yamagata lineages that cocirculated in the years following their birth, and individuals born in the early 1990s were likely exposed to the Yamagata lineages that predominated after they were born [4]. Sera from two individuals born in the 1980s yielded two different patterns; the M74 serum made the Victoria lineage B/CO virus more B/PK like and the F09 serum made the Yamagata lineage B/PK virus more B/CO like. Sera from two individuals born in the 1990s (F16 and M56) yielded escape viruses with similar changes, both making the Yamagata lineage B/PK virus more B/CO like. The trend observed with the latter sera is interesting. Even though they were vaccinated as young adults two years in a row with antigen from a Victoria lineage virus, B/Phuket escape viruses became more Victoria-like indicating a strong bias against Yamagata lineage viruses. This suggests that after vaccination there may have been recall and amplification of antibodies toward the lineage that predominated when they were children. These sera were also from individuals vaccinated with LAIV one year, followed by IIV the next. It is possible that the LAIV may have increased the abundance of cross-reactive antibodies as IBV infection in humans has been shown to induces cross-lineage HA stalk-specific antibodies [19]. The sample size is too small to draw any conclusions, but the results suggest that prior exposure to IBV affects the type of cross-reactive antibodies that are elicited after subsequent vaccination.

A concern recently raised is the possible detrimental effect on long-lasting protection against an H3N2 epitope induced by priming during childhood and vaccination with a mismatched strain [49]. However, for IBV it may be that the frequent circulation of strains from both lineages during childhood primes the immune system for both so that cross-reacting antibodies are available even if a mismatched strain is used in the vaccine. That said, Skonronski et al. observed lower vaccine effectiveness when the IBV vaccine component was not changed from year to year for patients ≥9 years old, regardless of the match to the circulating strains [50]. Our data set is too small to draw any conclusion but we did observe that the higher the titers were after repeated vaccination with the same antigen, the more likely changes were to occur in an escape virus.

There are several limitations with this work. Our study does not provide information about the mechanisms that gave rise to the apparent switching between lineages. We only performed these experiments with a limited number of sera samples and therefore did not perform statistical analyses. The study did include control sera from influenza naïve subjects. It appears that the diversity of the quasispecies contributed to the emergence of the escape viruses, but we do not know to what extent other factors such as the MOI, the type of antibody, or the fidelity of the virus replication played a role. The mutant viruses were not selected by continuous incubation in sera and this may have allowed virus not inhibited by antibodies during the inoculation to persist in in the population during the experiment. Finally, while in vitro experiments contribute to our scientific knowledgecare must be taken in applying the new knowledge to real world applications.

## 5. Conclusions

Here we have presented an analysis of the changes present in the HA gene of Yamagata and Victoria lineage IBVs after growth in the presence of human sera from vaccinated young adults. NGS data demonstrated that it is possible for the viral population from one lineage to change so much that it resembles that of the other lineage. This occurred with virus at a high multiplicity of infection after one to four passages on MDCK cells. The changes were observed in both the Yamagata to Victoria direction and vice versa. Additionally, identical changes were observed in known antigenic sites for viruses from both lineages that were grown in the presence of the same serum, indicating the presence of cross-reactive IBV antibodies. This provides some insight into the limits of mutation and antigenic drift in IBV and may prove useful for the development of universal vaccines targeting IBVs.

## Figures and Tables

**Figure 1 vaccines-08-00125-f001:**
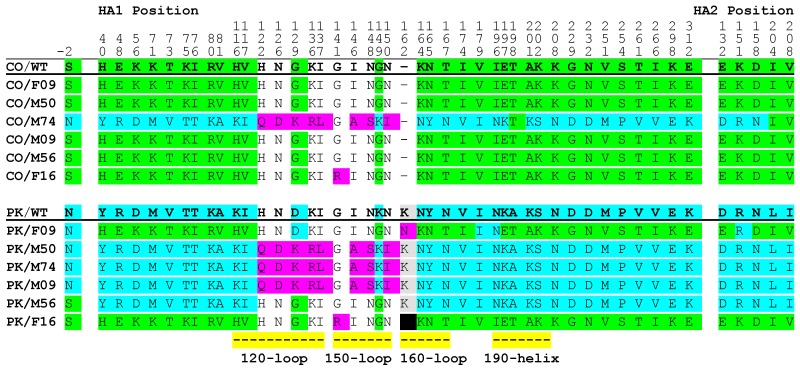
HA consensus (>50%) sequences of wild-type and escape viruses. Viruses are labeled as in Table 1. NGS sequence data for each escape virus was translated and aligned with the sequence from the input virus. The wild-type virus sequence is underlined an in bold font. Numbering is based on the Yamagata lineage B/Phuket/3073/2013 (PK/WT) sequence. Amino acid sequences common to the PK/WT sequence that differ from the B/Colorado/06/2017 (CO/WT) sequence are highlighted in blue, those common to CO/WT are highlighted in green; mutations resulting in an amino acid not present at levels greater than 50% in the wild-type viruses are shown in pink; and deletions gained through virus passage are highlighted in black. Amino acids associated with known antigenic regions are indicated by yellow bars.

**Figure 2 vaccines-08-00125-f002:**
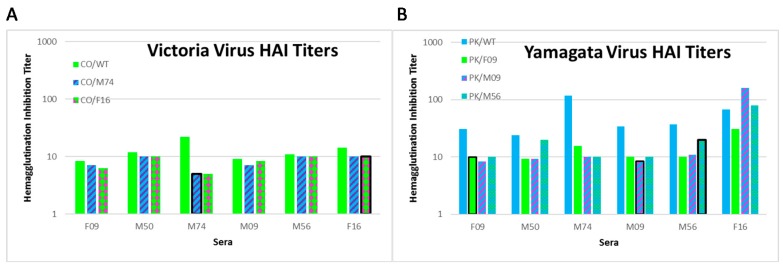
Hemagglutinin inhibition titer for select escape viruses. HAI titers for Victoria lineage viruses (**A**) and Yamagata lineage viruses (**B**) are shown. Viruses are labeled in the legend as for Table 1 and Table 2. The data for the CO/WT and PK/WT viruses (from Table 1 and Table 2) are shown as the green and blue bars in panels A and B, respectively. Graph bars are colored to indicate the predominance of B/CO-like (green) or B/PK-like (blue) amino acids at positions highlighted in Figure 1. Red stripes indicate changes to the 120- and 150-loops; and red dots indicate the G141R change. Graph bars with a black outline indicate that the virus was generated using the sera indicated in that column.

**Figure 3 vaccines-08-00125-f003:**
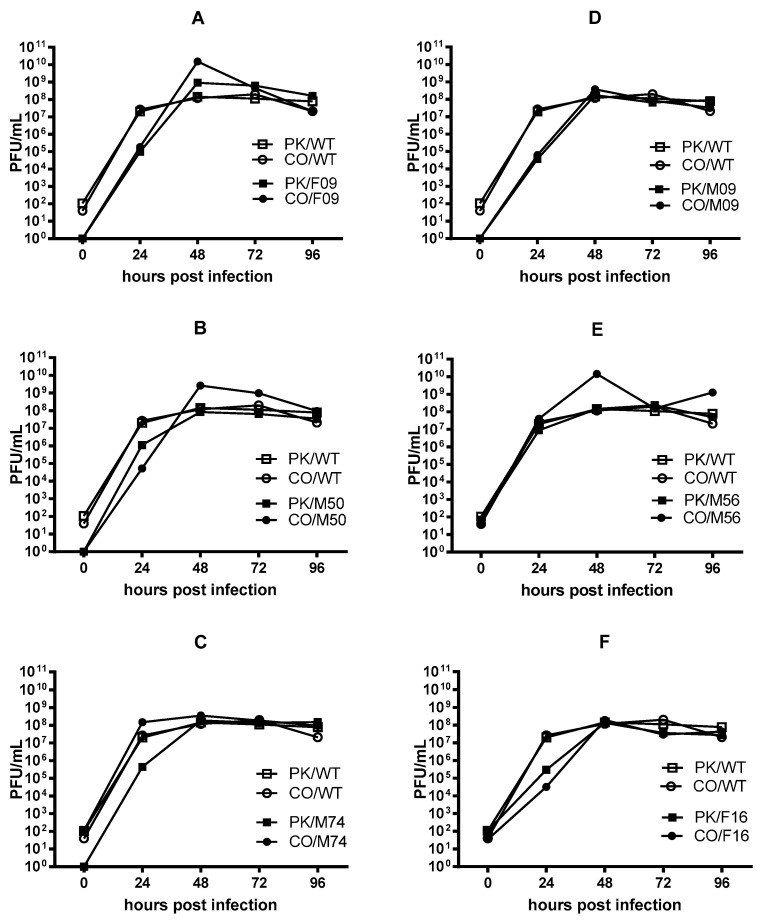
Multistep growth curves for escape viruses. MDCK cells were infected with a low MOI (0.001) of virus and the number of plaque forming units (PFU) present in the supernatant determined at various timepoints. The PK/WT and CO/WT growth curves are shown in each panel using open squares and circles respectively. The growth curves for the escape viruses are displayed as filled symbols. Growth curves for viruses generated using sera from the 2003 vaccine recipients are shown in panels (**A**–**C**) and growth curves for viruses generated using sera from the 2011 vaccine recipients are shown in panels (**D**–**F**).

**Table 1 vaccines-08-00125-t001:** Serum and Victoria lineage viruses. The designation for each post-vaccination serum (column 1) and year of vaccine receipt (column 2) are listed. Hemagglutination inhibition (HAI) titers toward IBV vaccine component (from [25]) are shown in columns 3 and 4. HAI titers (geometric mean from four replicates) for the sera toward the contemporary B/Colorado/06/2017 virus are in column 5. The number of passages used to attain the escape virus are shown and the name of each escape virus, which incorporates the serum name, are shown in the last two columns. HK330, B/Hong Kong/330/01; BR60, B/Brisbane/60/2008; CO_WT, B/Colorado/06/17; ND, not determined.

Serum	Years Vaccinated	HK330 HAI Titer	BR60 HAI Titer	CO_WT HAI Titer	Number of Passages	Escape Virus
**F09**	2002, 2003	10	ND	8	4	**CO/F09**
**M50**	2002, 2003	40	ND	12	4	**CO/M50**
**M74**	2002, 2003	320	ND	22	3	**CO/M74**
**M09**	2010, 2011	ND	10	9	4	**CO/M09**
**M56**	2010, 2011	ND	40	11	3	**CO/M56**
**F16**	2010, 2011	ND	320	14	1	**CO/F16**

**Table 2 vaccines-08-00125-t002:** Serum and Yamagata lineage virus. The designation for each post-vaccination serum (column 1) and year of vaccine receipt (column 2) are listed. HAI titers, the geometric mean from four replicates, toward the contemporary PK_WT (B/Phuket/3073/2013) virus are shown in column 3. The number of passages used to attain the escape virus and the name of each virus, which incorporates the serum name, are shown in the last two columns.

Serum	Years Vaccinated	PK_WT HAI Titer	Number of Passages	Escape Virus
**F09**	2002, 2003	31	4	**PK/F09**
**M50**	2002, 2003	24	3	**PK/M50**
**M74**	2002, 2003	118	3	**PK/M74**
**M09**	2010, 2011	34	4	**PK/M09**
**M56**	2010, 2011	37	4	**PK/M56**
**F16**	2010, 2011	67	3	**PK/F16**

**Table 3 vaccines-08-00125-t003:** Frequency of subpopulations in wild-type viruses encoding alternative sequence. The table provides information about the position of the codon (gene) and the amino acid excluding the signal peptide (protein) using the numbering for Yamagata virus B/Phuket/3073/2013 (PK-WT). The Victoria lineage B/Colorado/06/2017 (CO_WT) sequence has one less codon (codon position 177 and amino acid position 162). The consensus amino acids for PK_WT and CO_WT at the indicated positions are shown. The frequency of nucleotide variants that encode an alternative amino acid are shown in the adjacent columns. Unless otherwise indicated the most frequent alternative amino acid is that of the other virus lineage (shown in bold font). * two nucleotide changes required; ^#^ two nucleotides in codon changed at same frequency in escape mutants.

Gene	Protein	PK_WT	Yam-Vic	CO_WT	Vic-Yam
14	N/A	**Asn**	<0.05 ^#^	**Ser**	<0.05 ^#^
55	40	**Tyr**	0.39	**His**	<0.05
63	48	**Arg**	0.39 *^,#^	**Glu**	<0.05 *^,#^
71	56	**Asp**	0.39 *^,#^	**Lys**	<0.05 *^,#^
86	71	**Met**	0.36 *	**Lys**	<0.05 *
88	73	**Val**	0.37 *^,#^	**Thr**	<0.05 *^,#^
90	75	**Thr**	0.38 *^,#^	**Lys**	<0.05 *^,#^
91	76	**Thr**	0.38	**Ile**	<0.05
95	80	**Lys**	0.38 ^#^	**Arg**	<0.05 ^#^
96	81	**Ala**	0.35	**Val**	<0.05
131	116	**Lys**	0.46 *^,#^	**His**	<0.05 *^,#^
132	117	**Ile**	0.47	**Val**	<0.05
137	122	**His**	0.48 (Gln)	**His**	<0.05
141	126	**Asn**	0.47 (Asp)	**Asn**	<0.05
144	129	**Asp**	0.48 (Asn) ^#^	**Gly**	<0.05 ^#^
151	136	**Lys**	0.46 (Arg)	**Lys**	<0.05
152	137	**Ile**	0.47 (Leu)	**Ile**	<0.05
156	141	**Gly**	<0.05	**Gly**	<0.05
161	147	**Thr**	0.46 (Ala)	**Ile**	<0.05
164	149	**Lys**	0.49 *^,#^	**Gly**	<0.05 *^,#^
165	150	**Asn**	0.48 (Ile)	**Asn**	<0.05
179	164	**Asn**	0.45	**Lys**	<0.05
180	165	**Tyr**	0.45	**Asn**	<0.05
182	167	**Asn**	0.44 *^,#^	**Thr**	<0.05 *
189	174	**Val**	0.47 *^,#^	**Ile**	<0.05 *^,#^
194	179	**Ile**	<0.05	**Val**	<0.05
211	196	**Asn**	<0.05	**Ile**	0.23
212	197	**Lys**	0.36 ^#^	**Glu**	<0.05 *^,#^
213	198	**Ala**	0.47 *	**Thr**	0.07 *
216	201	**Lys**	0.36 *	**Ala**	<0.05 *^,#^
217	202	**Ser**	<0.05 *^,#^	**Lys**	<0.05 *^,#^
223	208	**Asn**	0.36	**Lys**	<0.05
245	230	**Asp**	0.35	**Gly**	<0.05
248	233	**Asp**	0.37	**Asn**	<0.05
267	252	**Met**	0.33	**Val**	<0.05
270	255	**Pro**	0.34	**Ser**	<0.05
277	262	**Val**	0.33 *^,#^	**Thr**	<0.05 *^,#^
282	267	**Val**	0.33	**Ile**	<0.05
314	299	**Glu**	0.34	**Lys**	<0.05
327	312	**Lys**	0.34	**Glu**	<0.05
493	478	**Asp**	0.42	**Glu**	<0.05
512	497	**Arg**	0.42 ^#^	**Lys**	<0.05 ^#^
519	504	**Asn**	0.40	**Asp**	<0.05
565	550	**Leu**	0.34	**Ile**	<0.05
569	554	**Ile**	0.34	**Val**	<0.05

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
