# Peer review of "Immune Pressure on Polymorphous Influenza B Populations Results in Diverse Hemagglutinin Escape Mutants and Lineage Switching"

_vaccines, 2020, doi:10.3390/vaccines8010125_

Round 1
Reviewer 1 Report
The authors present a nice study analyzing Influenza B virus escape mutations in the HA gene when the virus is in contact with sera from vaccinated subjects. They demonstrated in vitro that sera from vaccinated adults with an IBV from Victoria lineage contain cross-reactive antibodies allowing the rise of escape mutations in IBV viruses from both lineages, Victoria and Yamagata. NGS sequences data obtained from these experiments are very informative and helpful. The conclusions match the observations and the explanations provided by the authors. I include some questions as well as several minor items to address below.
Overall, english writing needs some improvement as some sentences are difficult for being understood by the reader.
The material and methods section needs to be improved with more details given on the experiments performed. Also, the different experiments described in the “Preparation of escape mutants” section need to be discriminated to help the reader.
In the “Results” section, the authors only describe the results they obtained from their experiments, it would be of interest to write a last sentence at the end of each paragraph describing each experiment with a clear conclusion explaining to which concepts are associated the results generated.
In figure 3, there is a typo error as “CO/M56” is written “C)/M56”.
In the figure.3, the graph E has a different police and size, please make it with the same settings than other graphs.
In figure 3, can the authors explain how the PFU/ml for the escape mutants at 0h post infection differs from the value of the WT viruses while the same MOI of 0.001 is used ?
In the discussion, p11 line 341, the authors conclude that conclude that 5 of the viruses had lower titers at 24h time point due to a possible loss of fitness before reaching similar titers at 72h post infection (Figure 3). I would agree with the escape virus F16 but it looks like for the 4 other escape viruses (F09, M50, M74 and M09), the growth curves are below WT at 24h post infection because it was already below at 0h post infection as the curves between 0h and 24h post infection follow parallel growth rate.
Author Response
Open Review 1
Comments and Suggestions for Authors
The authors present a nice study analyzing Influenza B virus escape mutations in the HA gene when the virus is in contact with sera from vaccinated subjects. They demonstrated in vitro that sera from vaccinated adults with an IBV from Victoria lineage contain cross-reactive antibodies allowing the rise of escape mutations in IBV viruses from both lineages, Victoria and Yamagata. NGS sequences data obtained from these experiments are very informative and helpful. The conclusions match the observations and the explanations provided by the authors. I include some questions as well as several minor items to address below.
Overall, english writing needs some improvement as some sentences are difficult for being understood by the reader.
We appreciate the reviewer’s concern and have tried to improve the flow of the manuscript through the changes suggested.
The material and methods section needs to be improved with more details given on the experiments performed. Also, the different experiments described in the “Preparation of escape mutants” section need to be discriminated to help the reader.
Concerns about the materials and methods section were raised by all three reviewers. We have expanded the description of experiments and added information about the cells and viruses used, and a description of the hemagglutination inhibition assays and geometric mean titer calculation (text on pages 2-3).
In the “Results” section, the authors only describe the results they obtained from their experiments, it would be of interest to write a last sentence at the end of each paragraph describing each experiment with a clear conclusion explaining to which concepts are associated the results generated.
We have added text describing the correlation between HAI titer toward vaccine antigen and contemporary viruses prior to Tables 1 and 2. We have noted (lines 198-201 before Figure 1) that the Victoria antigen viruses with the most changes were derived form experiments using sera with the highest titers in response to vaccination. We add that the 160 loop helps define the lineages but it remains unclear what role the potential glycosylations sites have (line 254). Prior to Figure 2 we note that our results show that the HAI titers are generally lower with the escape mutants indicating that fewer antibodies in the sera are able to bind. A sentence is added at line 309 stating that the changes to the viral population did not affect growth.
In figure 3, there is a typo error as “CO/M56” is written “C)/M56”.
The error has been corrected.
In the figure.3, the graph E has a different police and size, please make it with the same settings than other graphs.
This was also noted by reviewer 3 and the sizing has been corrected.
In figure 3, can the authors explain how the PFU/ml for the escape mutants at 0h post infection differs from the value of the WT viruses while the same MOI of 0.001 is used ?
Detecting infectious particles at the first timepoint post-infection indicates that some of the inoculum remained and that that virus was not bound to cells. A comment is included in line 305 on page 11.
In the discussion, p11 line 341, the authors conclude that conclude that 5 of the viruses had lower titers at 24h time point due to a possible loss of fitness before reaching similar titers at 72h post infection (Figure 3). I would agree with the escape virus F16 but it looks like for the 4 other escape viruses (F09, M50, M74 and M09), the growth curves are below WT at 24h post infection because it was already below at 0h post infection as the curves between 0h and 24h post infection follow parallel growth rate.
We have adopted the reviewer’s perspective and included it in the description of the time-courses on page 11. As stated above, the amount of virus detected o-hours post infection varied between the experiments indicating that some virus remained and was not attached to cells after the inoculum was removed. We report that some viruses grew at a rate parallel to the wild-type viruses regardless of the initial titer. The PK/F16 and CO/F16 viruses were the exception with lower titers at the 24-hour time-point and this is noted in line 310.
Reviewer 2 Report
A possible impetus for this manuscript is based on the cross-lineage immunological responses to Yamagata and Victoria lineages of influenza B virus, as reported by many investigators (Li et al. 2019, de Vries et al. 2018, Skowronski et al. 2012, and Hirano et al. 2018). To investigate the molecular basis of this phenomenon, the authors used post-vaccinated antisera to generate “escape” mutants, followed by analysis of their sequences by next-generation sequencing, which can detect minor virus populations. They found that there were common “compensatory mutations” in the “escape mutants”, and these common mutations might render antigenic similarities between these two lineages, hence “cross-reactive” immunity. The authors suggested using these common mutations and antigenic changes to design “universal” vaccine.
The rationale to use antisera to generate mutants is a reasonable one, however, the methodology and results presented in this manuscript have fatal flaws, as listed below:
Per described protocol, the selective pressure was by binding the antisera to the virus and layer onto MDCK cells, followed by washing. This is in contrast to most established protocols where the antisera is present throughout the cultivation of the virus, either in embryonated eggs or in MDCK cells. The authors' procedure only selected what was present in their stock virus, not subsequent escaped progeny viruses; Whereas the viruses were cultivated and selected in cell culture, but virus titration was by hemagglutination-inhibition. It’s not clear why the authors did not use neutralization test as a measure of “escape”; It is confusing and difficult to read Table 1 and Table 2. Regarding antisera, there was a high titer antiserum (M74) and low titer (F09 and ND?), with HAI of 320 versus 10, respectively; however, the other titers were presented in an uninterpretable numbers, e.g., 22 versus 8. Furthermore, it is not known why the authors didn’t focus on the high titer antiserum; in Fig. 1, CO/M74 is distinct from the rest—is it a misalignment? There was no statistics at all: with just three antisera and six “escaped mutants”, it is difficult to draw any conclusion, not mentioning that there was no statistical analysis at all; The discussion was unfocused, and was not substantiated by data
In short, this manuscript is incomplete. It should be rejected.
Author Response
Open Review 2
Comments and Suggestions for Authors
A possible impetus for this manuscript is based on the cross-lineage immunological responses to Yamagata and Victoria lineages of influenza B virus, as reported by many investigators (Li et al. 2019, de Vries et al. 2018, Skowronski et al. 2012, and Hirano et al. 2018). To investigate the molecular basis of this phenomenon, the authors used post-vaccinated antisera to generate “escape” mutants, followed by analysis of their sequences by next-generation sequencing, which can detect minor virus populations.
We thank the reviewer for bringing the Hirano paper to our attention. They reported three types of broadly reacting antibodies; those that bind Yamagata lineage viruses, those that bind both Yamagata and Victoria lineage viruses, and those that bind influenza A and B viruses. The predominance of Yamagata reactive antibodies compared to Victoria reactive antibodies correlates with the great expansion of Phuket (Yamagata lineage) subpopulations compared to the Colorado (Victoria lineage) subpopulations. We have added this reference in our discussion, page 16 line 460.
They found that there were common “compensatory mutations” in the “escape mutants”, and these common mutations might render antigenic similarities between these two lineages, hence “cross-reactive” immunity. The authors suggested using these common mutations and antigenic changes to design “universal” vaccine.
We understand that care is needed in the language used in the description of our work. We have described the emergence of a virus population that differs from the parental population and have used the phrase “escape mutants” to describe the variant positions in these populations because they were observed after virus was incubated with antibodies. We haven’t described the mutations as compensatory, where a new mutation has a beneficial effect in the presence of a deleterious mutation. We do note that many of the changes in the viral population occur in the same positions in viruses from two different lineages and, because reciprocal changes were observed between the lineages, we suggest that mutation of the genome may be limited to specific codons in some areas. While our observation is interesting from strictly a scientific perspective we would be remiss if we did not mention the practical vaccine applications such as the design of universal vaccines.
The rationale to use antisera to generate mutants is a reasonable one, however, the methodology and results presented in this manuscript have fatal flaws, as listed below:
Per described protocol, the selective pressure was by binding the antisera to the virus and layer onto MDCK cells, followed by washing. This is in contrast to most established protocols where the antisera is present throughout the cultivation of the virus, either in embryonated eggs or in MDCK cells.
We acknowledge that we have performed experiments using an unconventional approach. Unlike many experiments to generate escape mutants that use sera from animals vaccinated against the same strain, we used polyclonal sera from individuals vaccinated several years before the emergence of the viruses used in this work. In the real-world people are exposed to viruses that differ from the ones they have been vaccinated against but are, as the reviewer suggests, likely continuously exposed to antibodies during natural infection. We mixed different concentrations of sera with a fixed concentration of virus anticipating that only virus unencumbered by antibodies would be able to initiate a productive infection. It is likely that this process yielded a different response than growth in the continuous presence of antibody and because of this we present our work in the context of a viral population or quasispecies.
The authors' procedure only selected what was present in their stock virus, not subsequent escaped progeny viruses;
While it may be argued that our approach only amplified a subpopulation, there still has to be a reason for the emergence of that subpopulation and, accordingly, we use the terminology “escape mutant” or “escape virus” because the changes occurred after virus was incubated with antibodies. We do not shy away from the observation that the population arose from a subpopulation, and we do present our data in the context of a quasispecies.
Whereas the viruses were cultivated and selected in cell culture, but virus titration was by hemagglutination-inhibition. It’s not clear why the authors did not use neutralization test as a measure of “escape”;
After the virus and antibodies were incubated together and used to infected cells, this inoculum was removed. If CPE was detected after 24-48 hours we concluded that a new pool of virus had been produced. Titration established that viral replication had occurred. We subsequently measured the HAI titers of the sera using the escape population and compared that to the initial viral population (see figure 2). This indicated that the new viral populations bound less antibody supporting our inclusion of “escape” in our description.
It is confusing and difficult to read Table 1 and Table 2.
We did not adequately describe the first two tables and we apologize for this. We have added additional text on page 3, lines 164-7, to better describe data in Tables 1 and 2.
Regarding antisera, there was a high titer antiserum (M74) and low titer (F09 and ND?), with HAI of 320 versus 10, respectively; however, the other titers were presented in an uninterpretable numbers, e.g., 22 versus 8.
As indicated by all reviewers the methods section was inadequate. To address this concern we have expanded the description of experiments and added information about the cells and viruses used, and a description of the hemagglutination inhibition assays and geometric mean titer calculation (text on pages 2-3).
Furthermore, it is not known why the authors didn’t focus on the high titer antiserum;
We were initially interested in exploring whether subjects from different eras, when different lineages of influenza B virus circulated, had antibody repertoires that resulted in different escape populations. We were interested in finding our if the difference in titer contributed to the number of changes in the escape viruses. This is discussed on page 15, line 386. We have included a line in the description of sera on page 2, line 99 to introduce this.
in Fig. 1, CO/M74 is distinct from the rest—is it a misalignment?
It is not a misalignment. The raw data is available for others to analyze.
There was no statistics at all: with just three antisera and six “escaped mutants”, it is difficult to draw any conclusion, not mentioning that there was no statistical analysis at all;
We acknowledge that the sample size is small and it is difficult to draw conclusions (page 16, line 479). However, we do put forward reasons as to why the escape viruses may have been created in this work.
The discussion was unfocused, and was not substantiated by data
In short, this manuscript is incomplete. It should be rejected.
We describe an unusual observation and, in our attempt to provide a rationale explanation, we have tried to anticipate and address other perspectives.
Reviewer 3 Report
The manuscript by Plant et al presents a thorough analysis of influenza B virus escape mutants upon treatment with serum from patients that received a vaccination. Interestingly, this analysis describes an important observation based on the amino acid changes upon immunological pressures.
Major comments:
In the materials and methods the authors need to explain why a higher MOI was used for the Victoria lineage strains when compared to Yamagata lineage when performing the passages in MDCK cells. In addition, serum collected between 2-6 weeks post vaccination can vary in the quantity and quality of the antibodies present. The immunological pressure may be different among these samples. The authors should highlight these differences.
Another aspect that has not been addressed in this manuscript is the presence of the complement system. Antibodies can also be tested for their ability to fix and activate complement. Could this host response affect the changes observed?
Line 184: Substitutions from Glycine to Lysine and from Asp to Lysine in the same position have been previously described but it is not mentioned whether there is a biological significance. Any specific post-translational modification?
Figure 2B: it is not clear to the reader why the authors did not examine HA titers against the other escape mutants M50 and M74
Minor comments:
Line 103: indicate initials for inactivated influenza vaccine (IIV) since it is used in line 139. Same for “live attenuated influenza vaccine”
Figure 1: pink and red are hard to distinguish. Also highlight WT sequences for easy identification. Perhaps highlight or use arrows under the labels.
Figure 3: use same font size for all the graphs
Author Response
Open Review 3
Comments and Suggestions for Authors
The manuscript by Plant et al presents a thorough analysis of influenza B virus escape mutants upon treatment with serum from patients that received a vaccination. Interestingly, this analysis describes an important observation based on the amino acid changes upon immunological pressures.
Major comments:
In the materials and methods the authors need to explain why a higher MOI was used for the Victoria lineage strains when compared to Yamagata lineage when performing the passages in MDCK cells.
We initially anticipated that subjects vaccinated two years in a row with a Victoria lineage antigen would have more Victoria lineage antibodies and that a higher virus titer would be needed to have enough virus unbound by antibodies to initiate productive infection of the MDCK cells. We have added a sentence, page 3 line 113, stating why more B/Colorado virus was used.
In addition, serum collected between 2-6 weeks post vaccination can vary in the quantity and quality of the antibodies present. The immunological pressure may be different among these samples. The authors should highlight these differences.
We have added additional text in the discussion, page 15 line 392, highlighting this.
Another aspect that has not been addressed in this manuscript is the presence of the complement system. Antibodies can also be tested for their ability to fix and activate complement. Could this host response affect the changes observed?
The complement system is an important part of the immune system to control virus infections. Here the sera was heat inactivated so the changes in viral population should be just due to antibodies preventing infection. We have included that in the description of the sera, page 3 line 101, but do not further engage this discussion as the analyses are focused on the virus rather than the antibodies.
Line 184: Substitutions from Glycine to Lysine and from Asp to Lysine in the same position have been previously described but it is not mentioned whether there is a biological significance. Any specific post-translational modification?
The previous description (Horthongkham et al.,) did not allude to any biological significance and they reported changes to other amino acids as well. We have added that the changes in both lineages indicate the position is variable, page 7 line 238. There is much to learn about post-translational modifications in influenza proteins (Dawson & Mehle, 2018) and we have chosen to keep our discussion regarding post-translational modifications limited to the better characterized glycoslylation sites.
Figure 2B: it is not clear to the reader why the authors did not examine HA titers against the other escape mutants M50 and M74
We had limited amounts of sera and opted to get more robust data by performing more replicates of viruses that had substantial differences rather than those that were similar. Text has been added to line 259 to reflect this.
Minor comments:
Line 103: indicate initials for inactivated influenza vaccine (IIV) since it is used in line 139. Same for “live attenuated influenza vaccine”
The acronym has been added in line 96.
Figure 1: pink and red are hard to distinguish. Also highlight WT sequences for easy identification. Perhaps highlight or use arrows under the labels.
The figure has been modified to make it clearer. The WT sequences are now presented in underlined bold font and the red highlight has been changed to black.
Figure 3: use same font size for all the graphs
This was also noted by reviewer 1 and has been corrected.
Round 2
Reviewer 2 Report
It appears that the authors have addressed most of the concerns raised by the reviewers. I have no problem accepting the publication of this manuscript, with the following provisions:
- Add a final paragraph in the discussion regarding the potential pitfalls: a) the selection for mutants was not by continuous incubation with the antisera; b) small sample sample size, and hence no statistical analysis;
- Regarding CO/M74, the authors should include a supplementary file providing the sequence data (or the accession no. if deposited in a database);
- The tables should be reformated;
Author Response
We have addressed the reviewers comments in full as described below:
- Add a final paragraph in the discussion regarding the potential pitfalls: a) the selection for mutants was not by continuous incubation with the antisera; b) small sample sample size, and hence no statistical analysis;
We have added the following paragraph
“There are several limitations with this work. Our study does not provide information about the mechanisms that gave rise to the apparent switching between lineages. We only performed these experiments with a limited number of sera samples and therefore did not perform statistical analyses. The study did include control sera from influenza naïve subjects. It appears that the diversity of the quasispecies contributed to the emergence of the escape viruses, but we do not know to what extent other factors such as the MOI, the type of antibody, or the fidelity of the virus replication played a role. The mutant viruses were not selected by continuous incubation in sera and this may have allowed virus not inhibited by antibodies during the inoculation to persist in in the population during the experiment. Finally, while in vitro experiments contribute to our scientific knowledge care must be taken in applying the new knowledge to real world applications.”
- Regarding CO/M74, the authors should include a supplementary file providing the sequence data (or the accession no. if deposited in a database);
We have supplemented the accession information provided in the text with a supplementary file (referenced on line 138 in the materials section on page3) that provides the additional information about accessing the files.
- The tables should be reformated;
We have added additional text in the table legends stating what is present in each column to add clarity for the reader